# Image Interpolation with Regional Gradient Estimation

**Zuhang Jia** [1] and **Qingjiu Huang** [1,2,*]

1   School of Information and Electronic Engineering, Zhejiang Gongshang University, Hangzhou 310018, China;
    20020090032@pop.zjgsu.edu.cn
2   Control System Laboratory, Graduate School of Engineering, Kogakuin University, Tokyo 163-8677, Japan
*   Correspondence: huang@cc.kogakuin.ac.jp

**Abstract:** This paper proposes an image interpolation method with regional gradient estimation (GEI) to solve the problem of the nonlinear interpolation method not sufficiently considering non-edge pixels. First, the approach presented in this paper expanded on the edge diffusion idea used in CGI and proposed a regional gradient estimation strategy to improve the problem of gradient calculation in the CGI method. Next, the gradient value was used to determine whether a pixel was an edge pixel. Then, a 1D directional filter was employed to process edge pixels while interpolating non-edge pixels using a 2D directionless filter. Finally, we experimented with various representative interpolation methods for grayscale and color images, including the one presented in this paper, and compared them in terms of subjective results, objective criteria, and computational complexity. The experimental results showed that GEI performed better than the other methods in an experiment concerning the visual effect, objective criteria, and computational complexity.

**Keywords:** image interpolation; image enhancement; bicubic interpolation; nonlinear interpolation; image gradient; edge diffusion; regional gradient estimation

## 1. Introduction

Image scaling is an important element of image processing and is widely used in applications that include aviation, medicine, communication, meteorology, remote sensing, animation production, film composition, and the military [1–3]. Image quality can be improved by scaling images using hardware, but there is a cost. As a result, improving the software aspect, i.e., scaling digital images using interpolation techniques, is critical.

Many image scaling methods have been proposed recently [4–28], and these methods can be divided into two categories, one being sample-based super-resolution reconstruction [4–12] and the other being sample-free-based interpolation [13–28]. The main difference between the two is that sample-free-based interpolation uses mathematical methods to estimate pixels directly based on the known pixels, whereas super-resolution reconstruction requires training samples to establish a mapping relationship between low-resolution images and high-resolution images before it can use image block-matching and replacement to complete the interpolation.

Compared to super-resolution reconstruction, sample-free interpolation methods have lower time and space complexities. In terms of modeling characteristics, this type of method can be subdivided into linear interpolation methods and nonlinear interpolation methods. The representative methods of linear interpolation include nearest-neighbor interpolation, bilinear interpolation, and bicubic interpolation [3]. Linear interpolation methods do not consider the pixel's position to be interpolated during the interpolation process, which causes the blurring of the image's edges and prevents high-definition visual effects from being achieved.

Nonlinear interpolation methods mainly include methods based on wavelet transform [25–28] and methods based on edge information [13–24]. The interpolation is based on wavelet transform: first, wavelet transform is performed on the image; next, the classical

image interpolation method is used to interpolate the frequency domain coefficients; then, threshold processing is performed to obtain the required interpolation image. This method effectively combines the interpolation method and the band-pass filtering feature of the wavelet transform; thus, it can effectively maintain high-frequency detail in the image and improve the visual effect of image interpolation. The approach based on edge information uses a non-directional interpolation method to process non-edge pixels, while for edge pixels, a directional interpolation method is used according to the direction of the edge. This paper mainly studies the interpolation method based on edge information.

In general, image edges are regions where the gray level changes dramatically and its derivative exceeds a threshold, usually by only one or two pixels in width. Such pixels are used in 1D directional interpolation along the edge direction. Meanwhile, non-edge pixels are processed using linear interpolation. However, for non-edge pixels near the edge, if the neighborhood involved in their interpolation crosses the border, the boundaries are blurred after interpolation.

This paper proposes a nonlinear interpolation method called "image interpolation with regional gradient estimation". Compared to existing nonlinear interpretation methods, the GEI method can distinguish the edge pixels and non-edge pixels of an image more effectively. Specifically, the proposed method employs a regional gradient estimation strategy to reduce the error generated by the CGI method when judging the gradient of unknown pixels, thereby enhancing the image's interpolation effect.

First, the proposed method utilizes the CGI method's concept of edge diffusion to diffuse the image edges and determine the properties of the diffused image pixel. Second, the gradients of unknown pixels in high-resolution (HR) images are estimated using bicubic interpolation with regional gradients. The estimated gradient is then applied to determine the properties of the unknown pixel. According to the properties, different interpolation methods are selected, including 2D directionless interpolation for non-edge pixels and 1D directional interpolation for edge pixels. We experimented with various representative interpolation methods for grayscale and color images, including the one presented in this paper, and compared them in terms of subjective results, objective criteria, and computational complexity. The experimental results showed that GEI performed better than the other methods in an experiment concerning the visual effect, objective criteria, and computational complexity.

The structure of this paper is as follows. Section 1 is the introduction. Section 2 briefly describes the image interpolation problem and the idea of edge diffusion. Section 3 describes the principle and process of the proposed method. Section 4 mainly describes the experiment's design process and results. The paper is concluded in Section 5.

## 2. Fundamental Issues and Ideas

Image interpolation refers to estimating unknown high-resolution values based on known low-resolution pixels. We denote the low-resolution (LR) image as $I_{\text{LR}}(i, j)$, with a size of $M \times N$, and the high-resolution (HR) image with a size of $2M \times 2N$ as $I_{\text{HR}}(2i, 2j)$. Figure 1 shows the first step of the interpolation process and demonstrates the relationship between the $I_{\text{LR}}(i, j)$ pixels and the $I_{\text{HR}}(2i, 2j)$ pixels in the $2 \times 2$ image interpolation case. To obtain HR image pixel values, we duplicate the LR pixels at $(i, j)$ to $(2i - 1, 2j - 1)$ in the HR image, as shown in Equation (1). Next, we need to construct unknown pixels around known pixel points in $I_{\text{HR}}(2i, 2j)$.

$$I_{\text{LR}}(i, j) = I_{\text{HR}}(2i - 1, 2j - 1), 1 \leq i \leq M, 1 \leq j \leq N \tag{1}$$

To evaluate the interpolation method's effectiveness, we first downsample the original image; then, we interpolate the downsampled image and, finally, compare the interpolated image to the original image. Currently, the most common downsampling methods are direct-extraction downsampling and function downsampling. Direct-extraction downsampling means keeping the odd rows and columns while deleting the remaining pixels from the HR image; this process yields an LR image, as shown in Figure 1a. Its pixel

relationship is shown in Equation (1). Function downsampling makes use of MATLAB's "imresize" function, which includes three methods: "nearest", "linear", and "bicubic". When acquiring low-resolution images, direct-extraction downsampling is frequently used in the literature [18–21] on image interpolation methods.

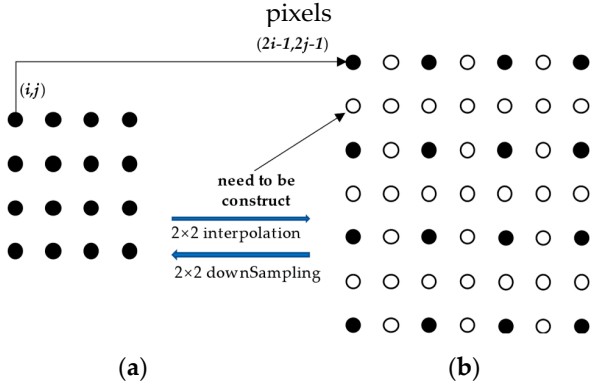

(a)                    (b)

**Figure 1.** Illustration of the interpolation. The dots denote the known pixels, and the circles denote the missing pixels. (**a**) low-resolution (LR) image; (**b**) high-resolution (HR) image.

The advantage of direct-extraction downsampling is that the obtained image pixels are part of the original image, allowing the original image information to be preserved to the greatest extent possible. This is consistent with the central question of image interpolation, that is, how to create an image as close to the original as possible with only pixels of the original image? The data obtained by function downsampling are calculated from the original image's pixel values, and this process destroys the original image's information. We performed bicubic interpolation on LR images obtained through various downsampling methods and used PSNR [29] to objectively evaluate the interpolated images. Table 1 summarizes the findings. We can see from Table 1 that the LR image obtained by direct extraction displayed superior PSNR performance after interpolation compared to the others, demonstrating that this method can preserve the greatest amount of original image information. Figure 2 depicts the results of the interpolation of the four downsampling methods used on the same test image, labeled "house". The luminance information of direct-extraction downsampling and nearest downsampling can be seen to be relatively complete. In contrast, the luminance information of bilinear downsampling and bicubic downsampling suffered from some loss. In Section 4, the low-resolution image is obtained by direct-extraction downsampling.

**Table 1.** A comparison of different downsampling methods with respect to the PSNR (dB).

| Method | Interlaced Row/Column | Nearest | Bilinear | Bicubic |
|---|---|---|---|---|
| PSNR | 31.13 | 27.56 | 26.09 | 25.17 |

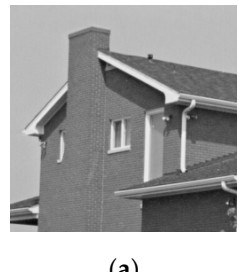   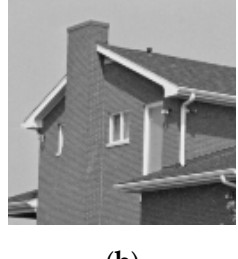   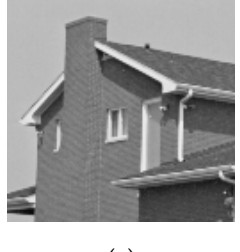   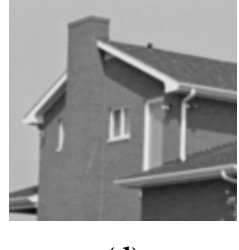   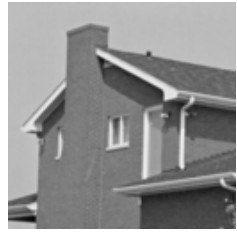

(a)             (b)             (c)             (d)             (e)

**Figure 2.** Interpolation results of different downsampling methods. (**a**) test image; (**b**) direct extraction; (**c**) nearest; (**d**) bilinear; (**e**) bicubic.

Figure 3a shows a relatively simple image, and Figure 3b shows the edges considered by the standard nonlinear image interpolation method. Point C in Figure 3b is located on the edge considered by the standard nonlinear image interpolation method, and the nonlinear interpolation method can ensure that the pixels on the edge appear clearly in the HR image. Because points A and B in Figure 3b are considered non-edge pixels, linear interpolation is used for them. However, the interpolation neighborhood for pixel B crosses the edge, and if the gray values on both sides of the edge change abruptly, the image edges will be blurred after nonlinear interpolation. To ensure the integrity of the interpolation effect, non-edge pixels within a specific range from the edge should be considered separately; thus, in this work, different interpolation strategies were employed for them. Figure 3c depicts the edge after edge diffusion. Because the edge after diffusion covers pixel B, a nonlinear interpolation method is employed to process it. The variational method of image diffusion is used in the CGI method to achieve edge diffusion [19]. This paper focuses solely on images after gradient edge diffusion has already been performed.

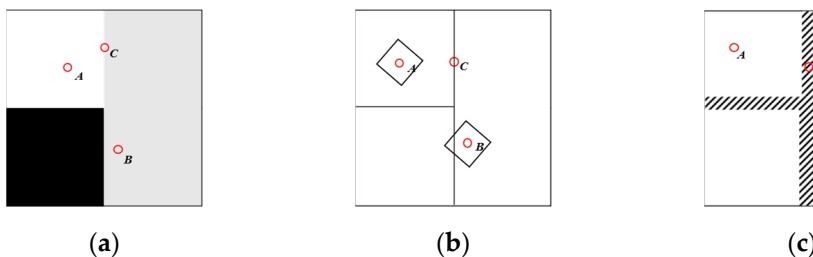

|       |       |       |
|:-----:|:-----:|:-----:|
| (**a**) | (**b**) | (**c**) |

**Figure 3.** Two different ways of interpreting the edges of an image. (**a**) test image; (**b**) canny edge; (**c**) edge using the CGI method.

Let us consider image interpolation at the pixel level. An image is made up of discrete pixel points. In Figure 4, the solid squares represent known pixels, and the hollow squares represent unknown pixels. After copying the pixels from the LR image into the HR image, there are three unknown pixels around $I_{HR}$ $(2i - 1, 2j - 1)$, namely the diagonal pixel at $(2i, 2j)$, the column pixel at $(2i, 2j - 1)$, and the row pixel at $(2i - 1, 2j)$. Image edges can only be formed along these four directions in the discrete state; we have simplified the possible edge directions to four cases, that is, $\theta = 0°$, $\theta = 45°$, $\theta = 90°$, and $\theta = 135°$, as shown in Figure 4.

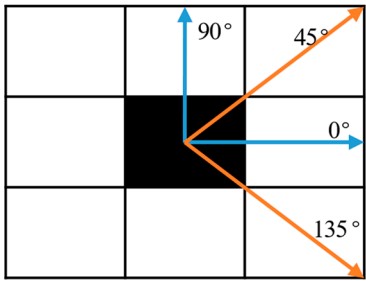

**Figure 4.** Illustration of edge directions in a digital image. The orange arrows represent the 45° and 135° directions to form a set of orthogonal groups, and the blue arrows represent the 90° and 0° directions to form a set of orthogonal groups.

It has been observed that variations in pixel intensity values along the tangent direction of the contrast boundary are always much smaller than those of the direction normal to the boundary. So, we can calculate the pixel gradient along two orthogonal directions to determine whether the pixel is located on the edge; if so, we need to determine the edge direction. Furthermore, if there is a significant difference between the two orthogonal directions, the pixel is classified as an edge pixel. To calculate the gradient difference, we must select a set of orthogonal edge directions. Let us choose 45° and 135° orthogonal

directions to determine whether the pixel is on a diagonal edge and 0° and 90° orthogonal directions to determine whether the pixel is on a horizontal or vertical edge.

We use four $3 \times 3$ convolution masks to calculate the gradient approximations in these four directions, as shown in the following formula, where $U_\theta$ denotes the value of the gradient value of $I_{LR}(i, j)$ in the $\theta$ direction.

$$U_0(i, j) = \left| \begin{bmatrix} 1 & 0 & 1 \\ 1 & 0 & 1 \\ 1 & 0 & 1 \end{bmatrix} * I_{LR}(i, j) \right| \tag{2}$$

$$U_{45}(i, j) = \left| \begin{bmatrix} 0 & 0 & 1 \\ -1 & 0 & 1 \\ -1 & -1 & 0 \end{bmatrix} * I_{LR}(i, j) \right| \tag{3}$$

$$U_{90}(i, j) = \left| \begin{bmatrix} 1 & 1 & 1 \\ 0 & 0 & 0 \\ -1 & -1 & -1 \end{bmatrix} * I_{LR}(i, j) \right| \tag{4}$$

$$U_{135}(i, j) = \left| \begin{bmatrix} 1 & 1 & 0 \\ 1 & 0 & -1 \\ 0 & -1 & -1 \end{bmatrix} * I_{LR}(i, j) \right| \tag{5}$$

Secondly, we need to assess the properties of the pixel in the LR image. Specifically, we need to judge whether a pixel is an edge pixel. If so, what is the direction of the edge on which it is located? The specific method of using the gradient to determine the properties of pixels is as follows.

As shown in Figure 1, for diagonal pixels, such as the points at the $(2i − 1, 2j − 1)$ and $(2i, 2j)$ pixel positions, if a pixel in the 45° or 135° direction satisfies $|u_{45°} − u_{135°}| \geq T$ ($u_\theta$ denotes the $\theta$-directional gradient for pixels), the pixel can be regarded as an edge point, and the direction of the edge on which the pixel is located can then be determined by Equation (6), where $u_{45}^{(h)}$ denotes the pixel's 45°-directional gradient in the HR image.

$$\begin{cases} \theta = 135^{\circ}, u_{45}^{(h)} \geq u_{135}^{(h)} \\ \theta = 45^{\circ}, \ u_{45}^{(h)} < u_{135}^{(h)} \end{cases} \tag{6}$$

Similarly, for a pixel in the horizontal or vertical direction, e.g., the pixels at the $(2i − 1, 2j)$ and $(2i, 2j − 1)$ positions, if the pixel gradient $u$ in the 0° or 90° direction satisfies $|u_{0°} − u_{90°}| \geq T$, the pixel can be determined as an edge point, and the edge where the pixel is located can be determined by Equation (7). According to the experiment in [19], the value of $T$ is about 0.01.

$$\begin{cases} \theta = 90^{\circ}, u_0^{(h)} \geq u_{90}^{(h)} \\ \theta = 0^{\circ}, \ u_0^{(h)} < u_{90}^{(h)} \end{cases} \tag{7}$$

To obtain the HR image, we first copy the pixel values and properties at the $(i, j)$ position in the LR image to the $(2i − 1, 2j − 1)$ position in the HR image, as shown in Equation (8).

$$I_{HR}(2i − 1, 2j − 1) = I_{LR}(i, j) \tag{8}$$

To interpolate the remaining unknown pixels in the HR image, we need to judge the properties of the unknown pixels according to the known pixel properties. As shown in Equation (8) [12], the CGI method uses the nearest-neighbor strategy to determine the properties of unknown pixels.

$$\begin{cases} u_\theta^{(h)}(2i − 1, 2j) = u_\theta^{(h)}(2i − 1, 2j − 1) \\ u_\theta^{(h)}(2i, 2j − 1) = u_\theta^{(h)}(2i − 1, 2j − 1) \\ u_\theta^{(h)}(2i, 2j) = u_\theta^{(h)}(2i − 1, 2j − 1) \end{cases} \tag{9}$$

The CGI method is not very precise in judging the properties. When the image's edge information is rich, this strategy may incorrectly judge the properties of unknown pixels, affecting the final interpolation effect. The arrows in Figure 5a represent the actual gradient directions of the a and b pixels in the test image. Because the CGI method uses the nearest-neighbor strategy, the gradient and direction of pixel b in Figure 5b are consistent with pixel a, and the direction is 135°. At this point, pixel b is interpolated as an edge pixel in the 135° direction. However, Figure 5a shows that the gradient direction of pixel b is 0°. So, we propose a region-based gradient estimation strategy to effectively judge the properties of pixels. The details are discussed in the following section.

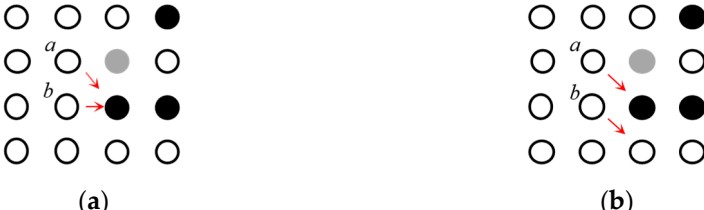

|  (a)  |  (b)  |

**Figure 5.** Comparison of local gradient calculation results. The red arrow represents the direction in which the pixel intensity value changes the most, which can also be called the gradient value. (**a**) the gradient of pixel a and pixel b in the original image; (**b**) the gradient calculation result of the CGI method.

## 3. Image Interpolation with Regional Gradient Estimation

### 3.1. Regional Gradient Estimation

We introduced a bicubic interpolation method to calculate the gradient value of unknown pixels. Bicubic interpolation uses the gradient values of 16 points around the point to be interpolated for cubic interpolation; this approach not only considers the influence of the gradient of the four directly adjacent points but also considers the influence of the gradient value change rate between adjacent points. However, if bicubic interpolation is performed on all unknown pixels in an image, the image's local characteristics are ignored, resulting in a large computational burden. In reality, images have both flat areas and areas with rich texture details. The image gradient changes smoothly in the flat area. For such areas, the effect obtained by a complex method is the same as that obtained by a simple method, but the calculation requirements increase significantly. Therefore, by using an interpolation method with lower computational requirements for flat areas and using bicubic interpolation in areas with intricate details, the amount of computation can be reduced while maintaining the quality of the enlarged image. Based on this consideration, we need to improve the bicubic interpolation method according to the region.

To obtain an HR image, as shown in Equation (8), we first copied the pixel values and properties at the $(i, j)$ position in the LR image to the $(2i - 1, 2j - 1)$ position in the HR image. Next, we proposed a gradient estimation strategy. As shown in Figure 6, for the pixel at position $(i + v, j + u)$, we calculated the gradient variance *Var* of the four-pixel in the $(i + v, j + u)$ position neighborhood in the original image. If the variance was less than the threshold T, the average of these four gradient values was taken as the gradient value of the pixel at $(i + v, j + u)$; otherwise, we used bicubic interpolation to calculate the gradient. Equation (10) is the formula for calculating the variance Var.

$$Var = (U - u_{11})^2 + (U - u_{12})^2 + (U - u_{21})^2 + (U - u_{22})^2 \qquad (10)$$

In Equation (11), $u_{11}$, $u_{12}$, $u_{21}$, and $u_{22}$ are the gradient values of the four pixels around position $(i + u, j + v)$ in the LR image, and $U$ is the mean value of the gradient of the four pixels. Calculating U requires one multiplication and three additions. From Equation (10), we need four multiplications and seven additions to calculate the variance. In total, five multiplications and ten additions are required, much less than the 70 multiplications and 45 additions required to calculate the pixel gradients by bicubic

interpolation. According to the experiments, the reduction in computation and the maintenance of image quality remain relatively reasonable when the threshold T = 18 [30]. Although the variance is calculated once for each image point, the total number of operations is still significantly reduced compared to bicubic interpolation.

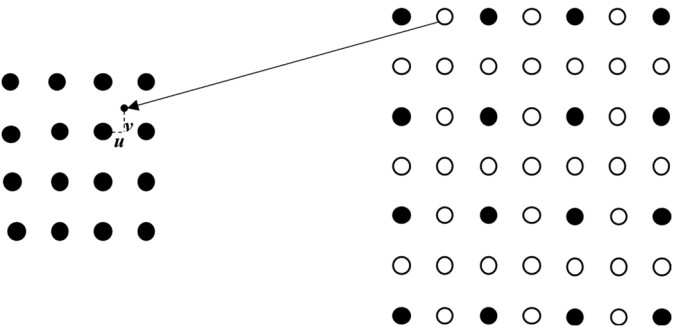

**Figure 6.** Bicubic interpolation mapping.

Figure 7 shows the visualization results of gradient calculations for the same test image, using both the method in this paper and the CGI method. Figure 7a shows the test image, and Figure 7b shows the result of the gradient visualization of the test image. The gradient visualization of the HR image that was estimated from the LR image using the CGI method is shown in Figure 7c. The result of the GEI method is shown in Figure 7d. The gradient calculated by the CGI method produced significant artifacts, especially at edge intersections, with significant errors when compared to the original image gradient, as shown in the yellow box in Figure 7c. In contrast, the gradient visualization estimated by the proposed method was close to the original image.

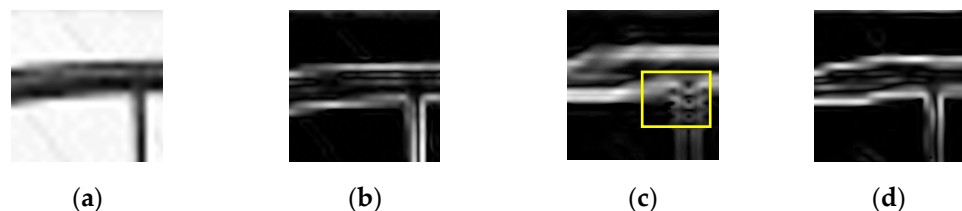

(**a**)             (**b**)             (**c**)             (**d**)

**Figure 7.** Comparison of visualization results for gradient estimation. (**a**) test image; (**b**) gradient calculation result of the test image; (**c**) gradient estimation result of the CGI method; (**d**) gradient estimation result of the GEI method.

### 3.2. Image Interpolation

After estimating the gradient of the unknown pixel in the high-resolution image, as mentioned in Section 2, we used the gradient to judge the properties of the unknown pixel and then calculate the pixel value according to the following steps. First, we calculated the gray value $I_{\text{HR}}$ (2$i$, 2$j$) at position (2$i$, 2$j$) in the HR image according to the gray value $I_{\text{LR}}$ ($i$, $j$) at position ($i$, $j$) in the LR image. Second, $I_{\text{HR}}$ (2$i$ − 1, 2$j$) and $I_{\text{HR}}$ (2$i$, 2$j$ − 1) were calculated from $I_{\text{LR}}$ ($i$, $j$) and $I_{\text{HR}}$ (2$i$, 2$j$), respectively. We adopted this order of calculation because the calculation of $I_{\text{HR}}$ (2$i$ − 1, 2$j$) and $I_{\text{HR}}$ (2$i$, 2$j$ − 1) requires $I_{\text{LR}}$ ($i$, $j$) and $I_{\text{HR}}$ (2$i$, 2$j$). Finally, each non-edge pixel was interpolated using bicubic interpolation. For edge pixels, we used Equation (11) for interpolation [20].

$$I = \omega(I_a + I_b) + (0.5 - \omega)(I_c + I_d) \qquad (11)$$

$\omega$ is an adjustable parameter. We set up an interpolation experiment to estimate the optimal value of $\omega$. Figure 8 shows PSNR results of our proposed method averaged for 12 test images using different values of $\omega$. In this work, we obtained the highest PSNR value when $\omega$ = 0.575; therefore, we regarded this value as the optimal value of $\omega$ and applied it to all our experiments.

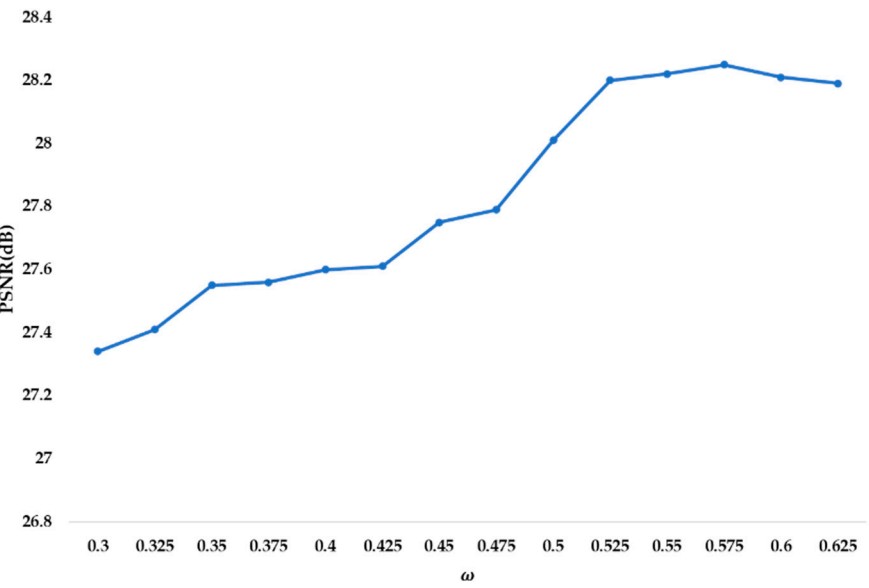

**Figure 8.** PSNR results averaged for 12 test images for different values of ω.

$I$ represents the gray value of the pixel to be interpolated; $I_a$, $I_b$, $I_c$, and $I_d$ represent four known pixel values around the unknown pixel, and their values depend on the properties of the unknown pixel. The specific process is as follows [20].

Let us first address pixels on the diagonal in the HR image. If $\theta = 135°$,

$$I_a = I_{HR}(i-1, j-1), I_b = I_{HR}(i+1, j+1) \tag{12}$$

$$I_c = I_{HR}(i-3, j-3), I_d = I_{HR}(i+3, j+3) \tag{13}$$

if $\theta = 45°$,

$$I_a = I_{HR}(i+1, j-1), I_b = I_{HR}(i-1, j+1) \tag{14}$$

$$I_c = I_{HR}(i+3, j-3), I_d = I_{HR}(i-3, j+3) \tag{15}$$

Let us now address pixels in the horizontal or vertical direction in the HR image. If $\theta = 90°$,

$$I_a = I_{HR}(i-1, j), I_b = I_{HR}(i+1, j) \tag{16}$$

$$I_c = I_{HR}(i-3, j), I_d = I_{HR}(i+3, j) \tag{17}$$

if $\theta = 0°$,

$$I_a = I_{HR}(i, j-1), I_b = I_{HR}(i, j+1) \tag{18}$$

$$I_c = I_{HR}(i, j-3), I_d = I_{HR}(i, j+3) \tag{19}$$

where $\theta$ is the edge direction of the edge where the pixel is located and can be derived from Equations (2)–(5).

So far, this paper has only addressed $2 \times 2$ interpolation. For example, when the image enlargement factor is four, $2 \times 2$ interpolation is performed on the original image twice. For the interpolation factor $M$ that is not a power of two, one can first apply $N \times N$ image interpolation (where $N$ is a power of two and smaller than but as close as possible to $M$), followed by 2D bicubic image interpolation with a rational number as the image enlargement factor. For example, if an image requires $9 \times 9$ image interpolation, we can first use the proposed method to interpolate the image by $8 \times 8$, followed by a $9/8 \times 9/8$ bicubic image interpolation.

Figure 9 shows the structure of the proposed method. First, we identified and diffused the LR image's edge pixels. Second, the gradient and pixel properties of the HR image were estimated. Finally, the pixels of the HR image were separated into edge and non-edge pixels. Multiple interpolation methods were then used to interpolate the classified pixels.

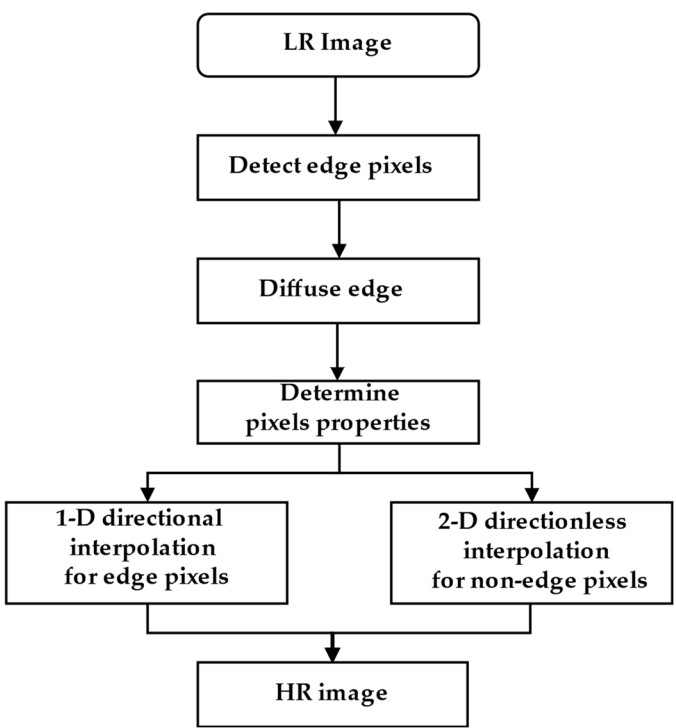

**Figure 9.** Structure of the proposed method.

## 4. Experiments

In order to evaluate the method presented in this paper, seven representative current methods were selected for experimentation: IBI [16], IEDI [17], CGI [18], CED [19], PGI [20], PCI [21], and WTCGI [25]. The source code of each method was either written by the method's proponent or implemented by the authors of this paper, and the individual parameter configurations used were the default parameters recommended by each method's proponent.

The proposed method (GEI) is based on the concept of CGI edge diffusion and uses CED to replace the iterative process in the CGI method to reduce time complexity. In contrast, to improve upon the CGI method, PCI and PGI employ the prediction-correction and gradient-prediction strategies, respectively. WTCGI combines wavelet transformation and the contrast-guided approach from CGI. Our experiment included the above five methods, as well as IEDI and IBI. IEDI is an improvement of the edge-direction interpolation method, and IBI is an improvement of the bicubic method, so comparing them to the method proposed in this paper is meaningful.

The twelve test images used in the image experiment are shown in Figure 10. The labels and sizes are as follows: cameraman (256 × 256 pixels), house (256 × 256 pixels), butterfly (512 × 521 pixels), bike (500 × 500 pixels), boats (512 × 512 pixels), wheel (512 × 512 pixels), airplane (256 × 256 pixels), stars (600 × 600 pixels), Barbara (512 × 512 pixels), fence (500 × 500 pixels), peppers (256 × 256 pixels), and baboon (256 × 256 pixels). These twelve images are frequently used in image processing and have a high reference value. We downsampled the reference images by direct extraction to generate LR images and then performed a 2 × 2 image interpolation on the LR images to generate HR images of the same size as the reference images, enabling us to compare the experimental outcomes of each approach.

Table 2 displays the PSNR results obtained from all of the interpolation methods under consideration, demonstrating that PCI and our proposed method outperformed all other methods. Furthermore, when comparing the methods for all of the test images, the average PSNR gain produced by the proposed method was close to that of PCI and better than all methods other than PCI. The proposed method provided the best PSNR perfor-

mance for images with sharper and more distinct edges, such as the test images "baboon" and "wheel".

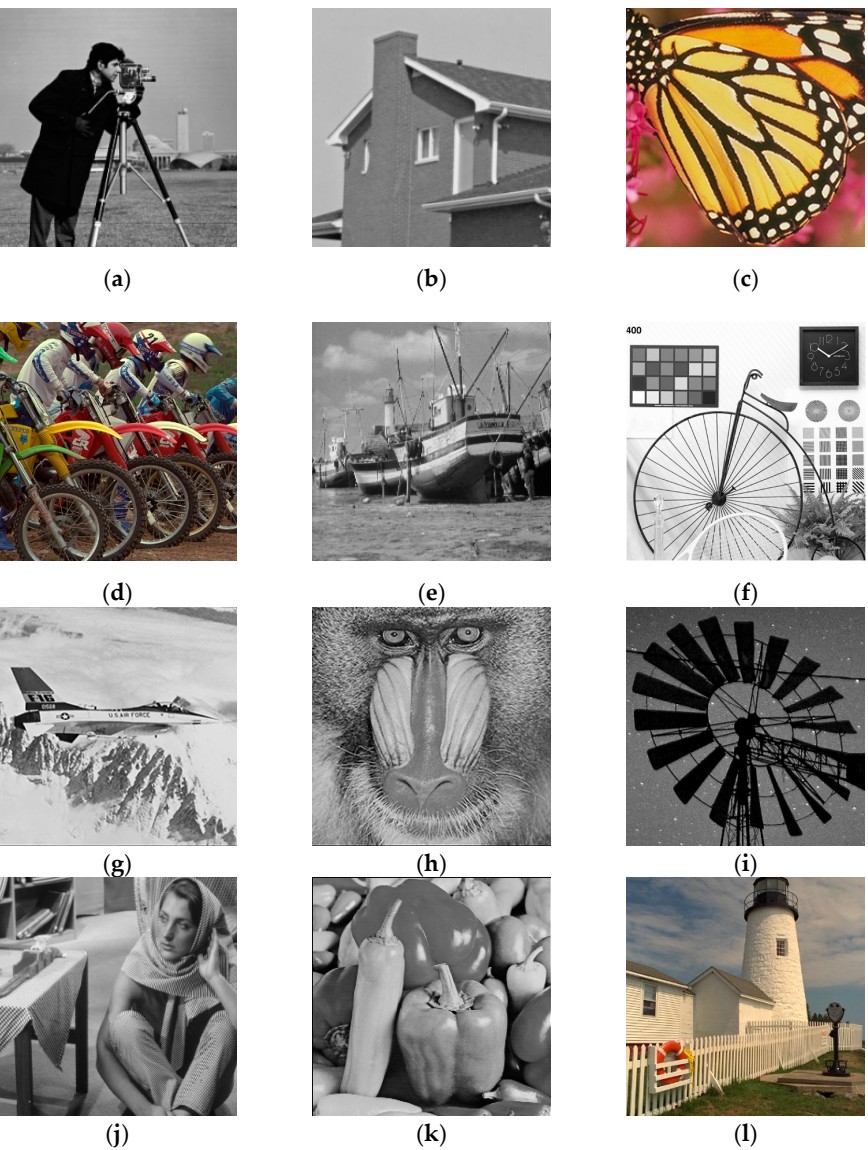

**Figure 10.** Test images. (**a**) cameraman; (**b**) house; (**c**) butterfly; (**d**) bike; (**e**) boats; (**f**) wheel; (**g**) airplane; (**h**) baboon; (**i**) stars; (**j**) barbara; (**k**) peppers; (**l**) fence.

Aside from the common practice of measuring the PSNR for objective performance evaluation, another image quality assessment metric known as structural similarity (SSIM) [29] was used to assess image quality because it correlates well with human visual perception. Table 3 reports the SSIM results (in dB) for all of the interpolation methods under consideration, demonstrating that the proposed method provided better SSIM performance for images with sharper and more distinct edges, such as the test images "butterfly", "wheel", and "baboon". Although the PCI method was significantly better than all other methods, the proposed method was close to PCI, CED and better than all methods other than PCI and CED in terms of average SSIM values.

The ability of a method to preserve the edges of an image is measured by the EPI (edge preservation index) [31]. Table 4 compares the edge preservation of each method when processing 12 images. The closer the EPI value is to 1, the better the method's edge preservation ability. The average EPI gain produced by the proposed method was better than other methods. Furthermore, the proposed method achieved the highest EPI values

for the images "cameraman", "Barbara", "butterfly", "baboon", "airplane", and "house". These results show that the proposed method was effective in preserving the edges of the images.

**Table 2.** A comparison of different interpolation methods with respect to the PSNR (dB).

| Test Images | CGI [18] | CED [19] | PCI [20] | IEDI [17] | PGI [21] | WTCGI [25] | IBI [16] | GEI (Proposed) |
|---|---|---|---|---|---|---|---|---|
| Bike | 25.82 | 25.82 | 25.90 | 25.17 | 25.92 | 25.21 | 25.32 | 25.85 |
| Wheel | 21.01 | 20.98 | 21.22 | 20.31 | 20.81 | 20.57 | 21.09 | 21.32 |
| Boats | 29.51 | 29.56 | 29.77 | 29.24 | 29.41 | 29.32 | 29.35 | 29.42 |
| Butterfly | 29.27 | 29.24 | 29.31 | 28.97 | 29.25 | 28.97 | 29.13 | 29.26 |
| House | 32.83 | 32.71 | 32.88 | 32.31 | 32.27 | 31.87 | 32.59 | 32.84 |
| Cameraman | 25.86 | 25.90 | 25.81 | 25.48 | 25.85 | 25.76 | 25.45 | 25.83 |
| Baboon | 22.50 | 22.41 | 22.53 | 22.41 | 22.51 | 22.35 | 22.34 | 22.59 |
| Peppers | 30.88 | 30.77 | 30.87 | 30.47 | 30.79 | 30.19 | 30.65 | 30.81 |
| Fence | 25.70 | 25.63 | 25.84 | 25.61 | 25.76 | 25.69 | 25.49 | 25.75 |
| Airplane | 26.54 | 26.49 | 26.59 | 26.60 | 26.52 | 26.10 | 26.43 | 26.61 |
| Barbara | 23.75 | 23.64 | 23.82 | 23.54 | 23.68 | 23.41 | 23.39 | 24.01 |
| Stars | 34.13 | 33.94 | 34.38 | 33.36 | 34.23 | 33.71 | 33.54 | 34.33 |
| Average | 27.32 | 27.26 | 27.41 | 26.96 | 27.25 | 26.93 | 27.06 | 27.39 |

**Table 3.** A comparison of different interpolation methods with respect to the SSIM (dB).

| Test Images | CGI [18] | CED [19] | PCI [20] | IEDI [17] | PGI [21] | WTCGI [25] | IBI [16] | GEI (Proposed) |
|---|---|---|---|---|---|---|---|---|
| Bike | 0.8808 | 0.8812 | 0.8803 | 0.8751 | 0.8785 | 0.8791 | 0.8783 | 0.8798 |
| Wheel | 0.8621 | 0.8626 | 0.8668 | 0.8644 | 0.8632 | 0.8649 | 0.8654 | 0.8665 |
| Boats | 0.8763 | 0.8801 | 0.8794 | 0.8771 | 0.8812 | 0.8744 | 0.8791 | 0.8796 |
| Butterfly | 0.9721 | 0.9732 | 0.9720 | 0.9718 | 0.9725 | 0.9698 | 0.9708 | 0.9758 |
| House | 0.8781 | 0.8778 | 0.8789 | 0.8783 | 0.8779 | 0.8775 | 0.8766 | 0.8780 |
| Cameraman | 0.8711 | 0.8732 | 0.8715 | 0.8704 | 0.8710 | 0.8692 | 0.8702 | 0.8732 |
| Baboon | 0.9125 | 0.9111 | 0.9130 | 0.9121 | 0.9174 | 0.9112 | 0.9114 | 0.9165 |
| Peppers | 0.9032 | 0.9041 | 0.9035 | 0.9029 | 0.8912 | 0.9026 | 0.9026 | 0.9025 |
| Fence | 0.7752 | 0.7780 | 0.7785 | 0.7763 | 0.7757 | 0.7765 | 0.7782 | 0.7723 |
| Airplane | 0.9405 | 0.9410 | 0.9401 | 0.9389 | 0.9438 | 0.9422 | 0.9409 | 0.9412 |
| Barbara | 0.9125 | 0.9128 | 0.9130 | 0.9114 | 0.9119 | 0.9105 | 0.9126 | 0.9118 |
| Stars | 0.9584 | 0.9603 | 0.9617 | 0.9608 | 0.9605 | 0.9610 | 0.9589 | 0.9608 |
| Average | 0.8952 | 0.8963 | 0.8966 | 0.8950 | 0.8954 | 0.8949 | 0.8954 | 0.8965 |

**Table 4.** A comparison of different interpolation methods with respect to the EPI (dB).

| Test Images | CGI [18] | CED [19] | PCI [20] | IEDI [17] | PGI [21] | WTCGI [25] | IBI [16] | GEI (Proposed) |
|---|---|---|---|---|---|---|---|---|
| Bike | 0.8258 | 0.8243 | 0.8325 | 0.8274 | 0.8313 | 0.8265 | 0.8236 | 0.8302 |
| Wheel | 0.8186 | 0.8189 | 0.8295 | 0.8111 | 0.8256 | 0.8274 | 0.8121 | 0.8259 |
| Boats | 0.7963 | 0.7865 | 0.7911 | 0.7895 | 0.7901 | 0.7898 | 0.7883 | 0.7916 |
| Butterfly | 0.8729 | **0.8775** | 0.8721 | 0.8703 | 0.8705 | 0.8749 | 0.8612 | 0.8753 |
| House | 0.7541 | 0.7533 | 0.7611 | 0.7615 | 0.7605 | 0.7596 | 0.7566 | 0.7624 |
| Cameraman | 0.7554 | 0.7525 | 0.7544 | 0.7494 | 0.7540 | 0.7556 | 0.7502 | 0.7601 |
| Baboon | 0.8029 | 0.8035 | 0.8051 | 0.8046 | 0.8069 | 0.8015 | 0.8014 | 0.8072 |
| Peppers | 0.7657 | 0.7654 | 0.7682 | 0.7625 | 0.7680 | 0.7652 | 0.7726 | 0.7677 |
| Fence | 0.7098 | 0.7085 | 0.7121 | 0.7074 | 0.7112 | 0.7098 | 0.7012 | 0.7091 |
| AirPlane | 0.8109 | 0.8112 | 0.8225 | 0.8112 | 0.8235 | 0.8165 | 0.8119 | 0.8254 |
| Barbara | 0.7450 | 0.7465 | 0.7519 | 0.7485 | 0.7515 | 0.7501 | 0.7526 | 0.7535 |
| Stars | 0.7983 | 0.7990 | 0.8013 | 0.7996 | 0.8012 | 0.8056 | 0.7989 | 0.8026 |
| Average | 0.7892 | 0.7873 | 0.7918 | 0.7869 | 0.7912 | 0.7902 | 0.7859 | 0.7926 |

The low computational complexity of our proposed method is an additional attractive feature. The average runtimes of each method for the interpolation of the 12 test images are documented in Table 5. Because the IBI method is based on the bicubic method, it does not have to consider image structure as much, making it the fastest. The CED method, which had the lowest runtime after IBI, replaces the edge-diffusion iterative process in the CGI method with thermal diffusion. Because of the inclusion of the regional gradient estimation strategy, the time complexity of the method proposed in this paper was slightly higher than that of PCI. IEDI was the slowest of all methods, with an average runtime nearly 11 times that of the method in this paper. The original author of the WTCGI method only provided C++ code, so it was not included in this comparison.

**Table 5.** Average runtime (in seconds) for interpolating 12 test images.

| CGI [18] | CED [19] | PCI [20] | IEDI [17] | PGI [21] | WTCGI [25] | IBI [16] | GEI (Proposed) |
|---|---|---|---|---|---|---|---|
| 3.10 | 0.89 | 1.31 | 17.96 | 1.54 | - | 0.17 | 1.45 |

For the subjective performance evaluation, we compared the performance of the methods in terms of visual effect. We conducted interpolation by $4 \times 4$ for the Wheel via 8 methods respectively, and select upper right part of "wheel" for subjective evaluation, as shown in Figure 11. The CGI and CED methods generated false edges at the junction edges, as shown in the yellow anchor boxes in Figure 11b,c; this was due to an ineffective gradient selection strategy, as shown in Equation (9). Although the CED method replaces the edge-diffusion iterative process in the CGI method with thermal diffusion, which increases running speed, false edges remain a problem. The PCI and PGI methods improve on the CGI method by using prediction correction and gradient prediction, respectively, to avoid the generation of false edges and produce sharper edges than the CGI method; however, their results yielded some edge loss in the red sight frames, as shown in Figure 11e,f. Although IEDE yielded more complete edges than the above-mentioned methods, 'speckle'-like noise was present, as shown in Figure 11g. WTCGI and IBI blurred more edges and generated more distinct artifacts, such as jagged edges, as seen in Figure 11h,i. The proposed method produced clear edges, avoided false and missing edges, and caused minimal artifacts, as shown in Figure 11d.

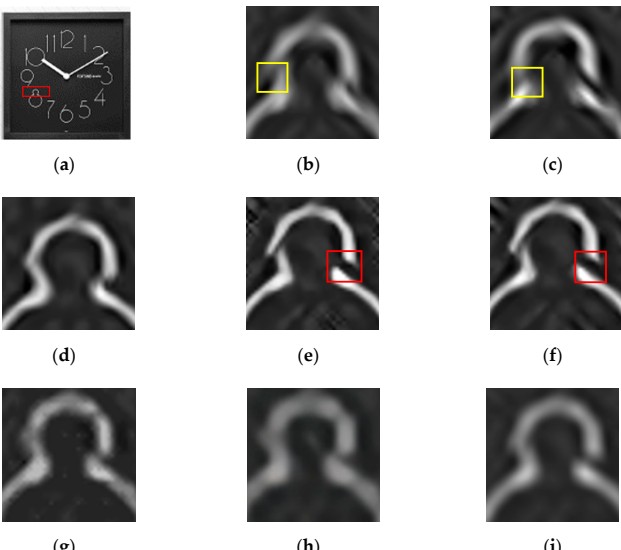

**Figure 11.** Comparison of interpolation results for top right part of the "wheel" image by different interpolation methods. (**a**) Upper right part of the "wheel" after interpolation; (**b**) CGI; (**c**) CED; (**d**) GEI; (**e**) PGI; (**f**) PCI; (**g**) IEDI; (**h**) WTCGI; (**i**) IBI.

For the color images, CGI and the proposed method were applied to interpolate the R, G, and B channels of the color images and combine them into a single image. Figure 12 shows the interpolation results obtained by the CGI method and the proposed method for the color images 'butterfly' and 'bike'. Figure 12b shows that the CGI method produced a ringing effect and artifacts in the local area. In contrast, GEI maintained the edges better in rich textural detail, as Figure 12c shows. Due to the introduction of a regional gradient estimation strategy, the proposed method had a slight advantage over the CGI method in interpolating color images.

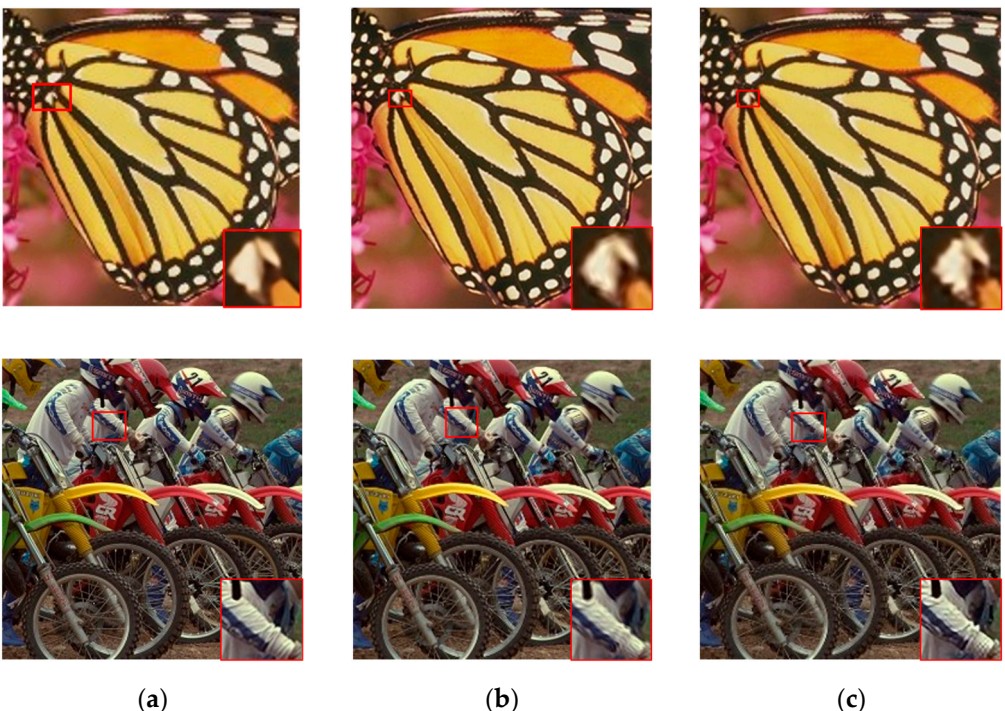

(a)       (b)       (c)

**Figure 12.** The comparison of color images interpolated by CGI and our method. (**a**) test image; (**b**) interpolation result by CGI; (**c**) interpolation result by GEI.

## 5. Conclusions

The processing of edges is essential for high-resolution image interpolation. The standard nonlinear interpolation methods tend to blur edges. To address the shortcomings of traditional nonlinear interpolation methods for non-edge pixels, this paper proposes an image interpolation method based on a region gradient estimation strategy.

To begin, the proposed method used the CGI method's edge diffusion idea to perform edge diffusion on low-resolution images. Then, this paper proposed a gradient estimation strategy to estimate the gradient of unknown pixels in high-resolution images to improve the shortcomings of the CGI in calculating the gradient, namely, using region-based bicubic interpolation to estimate the gradient of the pixel and judging its properties. This approach can determine the properties more effectively because this strategy takes into account the image's local characteristics. Following that, a 1D directional filter was used to process edge pixels, while a 2D directionless filter interpolated non-edge pixels. To validate the efficacy of the proposed method, experiments were conducted using a variety of representative interpolation methods for both grayscale and color images; we compared the results in terms of visual effect, objective criteria, and computational complexity. Extensive simulation results from both grayscale and color test images showed that our proposed image interpolation approach using regional gradient estimation outperformed various representative image interpolation methods with respect to both objective criteria and subjective visual quality assessments. Furthermore, when compared to other methods, the low computational complexity of the proposed method was a clear advantage.

**Author Contributions:** Conceptualization, Z.J.; methodology, Z.J.; software, Z.J.; validation, Z.J.; formal analysis, Z.J.; investigation, Z.J.; resources, Z.J.; data curation, Z.J.; writing—original draft preparation, Z.J.; writing—review and editing, Z.J. and Q.H.; visualization, Z.J. and Q.H.; supervision, Q.H.; project administration, Q.H. All authors have read and agreed to the published version of the manuscript.

**Funding:** This research received no external funding.

**Institutional Review Board Statement:** Not applicable.

**Informed Consent Statement:** The study did not involve humans.

**Data Availability Statement:** Data sharing is not applicable.

**Conflicts of Interest:** The authors declare no conflict of interest.

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
