# Peer review of "Image Interpolation with Regional Gradient Estimation"

_applsci, doi:10.3390/app12157359_

Round 1

Reviewer 1 Report

First of all, I would like to congratulate the authors for a well written manuscript. I have some issues to improve the quality of the final version (in my opinion):

1) In the equation (1) whydo you choose this formula? It means, there are a lot of posibilities to generate low resolution, so, why this one?

2) Why do you select those angles for edge direction? 

3) In equation (11) you say that omega=0,575 is the best parameter based on simulation results, what simulation? I think it is necessary to explain because this parameter is important.

4) The main problem, in my opinion, is the test images. I think that you have to imcrease the test dataset.

5) GENERAL: I think it is a good manuscript but it is necessary to specify your selections and if you propose a new method, it is necesary to test with a big dataset.

Reviewer 2 Report

The manuscript “Image Interpolation With Regional Gradient Estimation”, by Zuhang Jia and Qingjiu Huang, highlights an important topic in the image processing, meaning boundary recognition and processing. The manuscript can be published after minor revisions. My comments are bellow.

1.       Please insert more references in the Introduction section.

2.       Lines 67 – 71: Please correct the paragraph. You are discussing about chapters, but in your article do not appear such things. You have Sections/ sub-sections, not Chapters.

3.       Line 184: “…. Mentioned in Chapter 2..”. Again, please make corrections, you don’t have Chapters.

4.       Figure 7: Small comment: even the images are wide known as image tests, I have some concerns regarding Lenna image (Figure 7, d), which actually was published in Playboy magazine. Please take caution in the future, and try to avoid controversial photos.  As far as I know, the journals from IOP banned Lenna’s photo for appearing in scientific articles.

5.       How do you manage to resolve the problem of “parasitic” pixels, without damaging the resolution of your final result?

Reviewer 3 Report

An informed, according to regional gradient estimation, image interpolation method is presented in this article.

Experiments on selected cases demonstrate the superiority of the proposed approach over existing methods.

The paper is well structured and well presented.

The references and subjects of comparative are mostly outdated.

A considerable number of recent studies regarding image interpolation with edges can be found. They ought to be included in the literature review and the comparative study.

Round 2

Reviewer 1 Report

It´s ok in this version

Reviewer 3 Report

Issues in previous review had been addressed.